

# The geometry of Casimir W-algebras

Raphaël Belliard[1⋆], Bertrand Eynard[1,2†] and Sylvain Ribault[1‡]

**1** Institut de physique théorique, CEA, CNRS, Université Paris-Saclay
**2** CRM Montréal, QC, Canada

⋆ raphael.belliard@desy.de, † bertrand.eynard@ipht.fr, ‡ sylvain.ribault@ipht.fr

## Abstract

Let $\mathfrak{g}$ be a simply laced Lie algebra, $\widehat{\mathfrak{g}}_1$ the corresponding affine Lie algebra at level one, and $\mathcal{W}(\mathfrak{g})$ the corresponding Casimir W-algebra. We consider $\mathcal{W}(\mathfrak{g})$-symmetric conformal field theory on the Riemann sphere. To a number of $\mathcal{W}(\mathfrak{g})$-primary fields, we associate a Fuchsian differential system. We compute correlation functions of $\widehat{\mathfrak{g}}_1$-currents in terms of solutions of that system, and construct the bundle where these objects live. We argue that cycles on that bundle correspond to parameters of the conformal blocks of the W-algebra, equivalently to moduli of the Fuchsian system.



# 1 Introduction

In recent years, it has been found that two-dimensional conformal field theory with the central charge $c = 1$ can be formulated in terms of Fuchsian differential systems. In particular, this has led to the expression of Virasoro conformal blocks in terms of solutions of Painlevé equations [1], and to a new derivation of the three-point structure constant of Liouville theory [2].

A natural generalization would be that Fuchsian differential systems associated to a Lie algebra $\mathfrak{g}$ provide a formulation of conformal field theory based on the corresponding W-algebra $\mathcal{W}(\mathfrak{g})$ with the central charge $c = \operatorname{rank} \mathfrak{g}$. The Virasoro algebra with $c = 1$ would then be the special case $\mathfrak{g} = \mathfrak{sl}_2$. We call a W-algebra with the central charge $c = \operatorname{rank} \mathfrak{g}$ a Casimir W-algebra, because it coincides with the Casimir subalgebra of the affine Lie algebra $\widehat{\mathfrak{g}}_1$ at level one [3]. (For generic central charges, that Casimir subalgebra is much larger than $\mathcal{W}(\mathfrak{g})$, except in the special case $\mathfrak{g} = \mathfrak{sl}_2$.) This natural generalization has been illustrated by the analysis of conformal blocks in the case $\mathfrak{g} = \mathfrak{sl}_N$ [4]. And if Fuchsian systems describe $\mathcal{W}(\mathfrak{g})$ conformal blocks, then presumably they can also describe the corresponding generalizations of Liouville theory, namely conformal Toda theories.

Our motivation for investigating this natural generalization is to gain a structural understanding of two-dimensional CFT, which could be the basis for new computational techniques and for finding non-trivial duality relations. Of course, the interesting applications would mostly be confined to the case $\mathfrak{g} = \mathfrak{sl}_2$ of the Virasoro algebra, since most interesting CFTs have no larger W-algebra symmetry. But the generalization to W-algebras is essential for understanding geometrical structures, which were not discovered in previous work on the Virasoro case [2].

That previous work nevertheless gave us our starting point: the idea that the relation between Fuchsian systems and conformal field theory is naturally expressed in terms of objects that we will call amplitudes. Following [5], we will define amplitudes in terms of solutions of a differential system

$$\frac{\partial}{\partial x} M = [A, M] \,, \tag{1}$$

where $A$ and $M$ are $\mathfrak{g}$-valued functions of a complex variable $x$. In particular, by definition of a Fuchsian system, $A$ is a meromorphic function of $x$ with finitely many simple poles $z_1, \ldots, z_N$, and such that $A(x) \underset{x \to \infty}{=} O(\frac{1}{x^2})$. In conformal field theory, the amplitudes correspond to correlation functions of $N$ primary fields at $z_1, \ldots, z_N$, with additional insertions of chiral fields that we will call currents.

Our main aim in this work is to explore the intrinsic geometry of amplitudes. We will show that the relevant object is neither the Riemann sphere minus the singularities $\Sigma = \bar{\mathbb{C}} - \{z_1, \ldots, z_N\}$, nor even its universal cover $\widetilde{\Sigma}$, but a bundle $\widehat{\Sigma}$ that depends on the Lie algebra $\mathfrak{g}$ and on the function $A$. We will study cycles of $\widehat{\Sigma}$, and conjecture a relation between correlation functions and integrals of a certain form over certain cycles.

On the conformal field theory side, we will provide an interpretation of amplitudes and currents in terms of the affine Lie algebra $\widehat{\mathfrak{g}}_1$ at level one. In contrast to $\mathcal{W}(\mathfrak{g})$, the algebra $\widehat{\mathfrak{g}}_1$ is not a symmetry algebra of the theory, and our fields are not primary with respect to $\widehat{\mathfrak{g}}_1$. This is the reason why the amplitudes are not single-valued on $\Sigma$, and actually live on $\widehat{\Sigma}$.

## 2 The bundle and its amplitudes

### 2.1 Definition of the bundle

Assuming we know the $\mathfrak{g}$-valued function $A$, we first consider the Fuchsian differential system

$$\frac{\partial}{\partial x}\Psi = A\Psi \,, \tag{2}$$

where $\Psi$ is a function that takes values in a reductive complex Lie group $G$ whose Lie algebra is $\mathfrak{g}$. In other words, $\Psi$ is a section of a principal $G$-bundle over $\Sigma$ with the connection $d - A dx$.

As a $G$-valued function, $\Psi$ has nontrivial monodromies, and therefore lives on the universal cover $\widetilde{\Sigma}$ of $\Sigma = \bar{\mathbb{C}} - \{z_1, \ldots, z_N\}$. Given a closed path $\gamma \subset \Sigma$ that begins and ends at a point $x_0 \in \gamma$, the monodromy

$$S_\gamma = \Psi(x_0)^{-1}\Psi(x_0 + \gamma) \in G \,, \tag{3}$$

of $\Psi$ along $\gamma$ is in general nontrivial, and it is invariant under homotopic deformations of $\gamma$. The monodromy depends neither on the choice of $x_0 \in \gamma$, nor on the positions of the poles $z_j$, so $x_0$ and $z_j$ are isomonodromic parameters. The monodromy only depends on the residues of $A$ at its poles.

In terms of a given solution $\Psi$ of eq. (2), the solutions of the equation (1) can be written as $M = \Psi E \Psi^{-1}$, where $E \in \mathfrak{g}$ is constant $\frac{\partial E}{\partial x} = 0$, and $\Psi E \Psi^{-1}$ denotes the adjoint action of $\Psi$ on $E$. The idea is now to consider $M$ as a function not only of $x \in \widetilde{\Sigma}$, but also of $E$,

$$M(x.E) = \Psi(x) E \Psi(x)^{-1} \,. \tag{4}$$

While the pair $x.E$ apparently belongs to $\widetilde{\Sigma} \times \mathfrak{g}$, the monodromy of $\Psi$ along a closed path $\gamma$ can be compensated by a conjugation of $E$, so that

$$M((x + \gamma).(S_\gamma^{-1} E S_\gamma)) = M(x.E) \,. \tag{5}$$

Therefore, $M$ actually lives on the manifold

$$\widehat{\Sigma} = \frac{\widetilde{\Sigma} \times \mathfrak{g}}{\pi_1(\Sigma)} \,, \tag{6}$$

where the fundamental group of $\Sigma = \bar{\mathbb{C}} - \{z_1, \ldots, z_N\}$ acts on $\widetilde{\Sigma} \times \mathfrak{g}$ by (5). For $X = [x.E] \in \widehat{\Sigma}$, we call $\pi(X) = \pi(x)$ its projection on $\Sigma$.

Although our bundle $\widehat{\Sigma}$ has complex dimension $1 + \dim \mathfrak{g}$, we will only consider functions of $X = [x.E]$ that depend linearly on $E$, and the space of linear functions on $\mathfrak{g}$ is finite-dimensional. So the space of functions on $\widehat{\Sigma}$ that we will consider has the same dimension as the space of functions on a discrete cover of $\Sigma$. When viewed as bundles over $\Sigma$, the main difference between $\widehat{\Sigma}$ and a discrete cover such as $\widetilde{\Sigma}$ is that the fibers of $\widehat{\Sigma}$ have a linear structure: they are like discrete sets whose elements could be linearly combined.

### 2.2 Amplitudes

We assume that $\mathfrak{g}$ is semisimple, and write its Killing form as $\langle E, E' \rangle = \operatorname{Tr} E E'$, where the trace is taken in the adjoint representation. (Taking traces in an arbitrary faithful representation would not change the properties of amplitudes, but would kill the relation with conformal field theory that we will see in Section 3.)

For $X_1, \ldots, X_n \in \widehat{\Sigma}$ whose projections $\pi(X_i)$ are all distinct, let us define the connected and disconnected $n$-point amplitudes $W_n(X_1, \ldots X_n)$ and $\widehat{W}_n(X_1, \ldots X_n)$ as [5]

$$W_n(X_1, \ldots, X_n) = (-1)^{n+1} \sum_{\sigma \in \mathfrak{S}_n^{\text{circular}}} \text{Tr} \prod_{i \in \{1,\ldots,n\}} \frac{M(X_i)}{\pi(X_i) - \pi(X_{\sigma(i)})}, \tag{7}$$

$$\widehat{W}_n(X_1, \ldots, X_n) = \sum_{\sigma \in \mathfrak{S}_n} (-1)^\sigma \prod_{c \text{ cycle of } \sigma} \text{Tr} \prod_{i \in c} \frac{M(X_i)}{\pi(X_i) - \pi(X_{\sigma(i)})}, \tag{8}$$

where $\mathfrak{S}_n^{\text{circular}}$ is the set of permutations that have only one cycle of length $n$, and the factors in the products over $i$ are ordered as $(i_0, \sigma(i_0), \sigma^2(i_0), \ldots)$. For cycles of length one, we define $\text{Tr} \frac{M(X)}{\pi(X) - \pi(X)} = \text{Tr} A(\pi(X)) M(X)$, so that in particular

$$W_1(X_1) = \widehat{W}_1(X_1) = \text{Tr} A(\pi(X_1)) M(X_1). \tag{9}$$

Writing for short $W(\{i_1, \ldots, i_k\}) = W_k(X_{i_1}, \ldots, X_{i_k})$, the disconnected amplitudes are expressed in terms of connected amplitudes as

$$\widehat{W}(\{1, \ldots, n\}) = \sum_{\text{partitions } \sqcup_k I_k = \{1,\ldots,n\}} \prod_k W(I_k), \tag{10}$$

for example $\widehat{W}_2(X_1, X_2) = W_2(X_1, X_2) + W_1(X_1) W_1(X_2)$.

Let us study the behaviour of amplitudes near their singularities at coinciding points. To do this, it is convenient to rewrite the amplitudes in terms of the kernel

$$K(x, y) \underset{\pi(y) \neq \pi(x)}{=} \frac{\Psi(x)^{-1} \Psi(y)}{\pi(y) - \pi(x)}, \tag{11}$$

where $x, y \in \widetilde{\Sigma}$. For two points whose projections on $\Sigma$ coincide, we define the regularized kernel

$$K(x, y) \underset{\pi(x) = \pi(y)}{=} \Psi(x)^{-1} A(\pi(x)) \Psi(y) = \lim_{y' \to y} \left( K(x, y') - \frac{\Psi(x)^{-1} \Psi(y)}{\pi(y') - \pi(x)} \right). \tag{12}$$

For $X_i = [x_i . E_i]$, the amplitudes can then be written as

$$W_n(X_1, \ldots, X_n) = (-1)^{n+1} \sum_{\sigma \in \mathfrak{S}_n^{\text{circular}}} \text{Tr} \prod_{i \in \{1,\ldots,n\}} E_i K(x_i, x_{\sigma(i)}), \tag{13}$$

$$\widehat{W}_n(X_1, \ldots, X_n) = \sum_{\sigma \in \mathfrak{S}_n} (-1)^\sigma \prod_{c \text{ cycle of } \sigma} \text{Tr} \prod_{i \in c} E_i K(x_i, x_{\sigma(i)}). \tag{14}$$

Using our regularization (12), these expressions actually make sense even if some points $X_i$ have coinciding projections $\pi(X_i)$ on $\Sigma$, and we take these expressions as definitions of amplitudes at coinciding points. We can now write the behaviour of amplitudes in the limit $\pi(X_1) \to \pi(X_2)$. Choosing representatives $X_i = [x_i . E_i]$ such that $x_1 \to x_2$, we find

$$\widehat{W}_n(x_1 . E_1, x_2 . E_2, \ldots) \underset{x_1 \to x_2}{=} \frac{\langle E_1, E_2 \rangle}{x_{12}^2} \widehat{W}_{n-2}(\ldots) + \frac{1}{x_{12}} \widehat{W}_{n-1}(x_2 . [E_1, E_2], \ldots) + O(1), \tag{15}$$

where we used the notation $x_{12} = \pi(x_1) - \pi(x_2)$.

Let us study how amplitudes behave near the poles of $A(x)$. To the pole $z_j$, we associate $A_j \in \mathfrak{g}$ and $\Psi_j \in G$ such that

$$A(x) \underset{x \to z_j}{=} \frac{A_j}{x - z_j} + O(1), \tag{16}$$

$$\Psi(x) \underset{x \to z_j}{=} \Big( \mathrm{Id} + O(x - z_j) \Big)(x - z_j)^{A_j} \Psi_j, \tag{17}$$

$$M(x.E) \underset{x \to z_j}{\sim} (x - z_j)^{A_j} \Psi_j E \Psi_j^{-1} (x - z_j)^{-A_j}, \tag{18}$$

and the monodromy of $\Psi(x)$ around $z_j$ is

$$S_j = \Psi_j^{-1} e^{2\pi i A_j} \Psi_j. \tag{19}$$

Assuming that $A_j$ is a generic element of $\mathfrak{g}$, its commutant is a Cartan subalgebra that we call $\mathfrak{h}_j$. Let $R_j$ be the corresponding set of roots of $\mathfrak{g} = \mathfrak{h}_j \oplus \bigoplus_{r \in R_j} \mathfrak{g}_r$, and let us write the corresponding decomposition of $\Psi_j E \Psi_j^{-1}$ as

$$\Psi_j E \Psi_j^{-1} = E_j + \sum_{r \in R_j} E_r. \tag{20}$$

This allows us to rewrite the behaviour of $M(x.E)$ as

$$M(x.E) \underset{x \to z_j}{\sim} E_j + \sum_{r \in R_j} (x - z_j)^{r(A_j)} E_r. \tag{21}$$

Using $\langle A_j, E_r \rangle = 0$, we deduce

$$W_1(x.E) \underset{x \to z_j}{=} \frac{\langle A_j, E_j \rangle}{x - z_j} \big( 1 + O(x - z_j) \big) + \sum_{r \in R_j} (x - z_j)^{r(A_j)} \times O(1), \tag{22}$$

and more generally

$$\widehat{W}_n(x.E, \dots) \underset{x \to z_j}{\sim} \frac{\langle A_j, E_j \rangle}{x - z_j} \Big( \widehat{W}_{n-1}(\dots) + O(x - z_j) \Big) + \sum_{r \in R_j} (x - z_j)^{r(A_j)} \times O(1). \tag{23}$$

### 2.3 Casimir elements and meromorphic amplitudes

Let $\{e_a\}$ be a basis of $\mathfrak{g}$ and $\{e^a\}$ the dual basis such that $\langle e_a, e^b \rangle = \delta_a^b$. The center of the universal enveloping algebra $U(\mathfrak{g})$ is generated by Casimir elements of the type

$$C_i = \sum_{a_1, \dots, a_i} c_{a_1, \dots, a_i} e^{a_1} \otimes \cdots \otimes e^{a_i}, \tag{24}$$

where the rank $i$ of the invariant tensor $c_{a_1, \dots, a_i}$ is called the degree of $C_i$. In particular, the quadratic Casimir element is

$$C_2 = \sum_a e_a \otimes e^a. \tag{25}$$

For $x \in \widetilde{\Sigma}$, we define an amplitude that involves the Casimir element $C_i$ by

$$\widehat{W}_{i+n}(C_i(x), X_1, \dots, X_n) = \sum_{a_1, \dots, a_i} c_{a_1, \dots, a_i} \widehat{W}_{i+n}(x.e^{a_1}, \dots, x.e^{a_i}, X_1, \dots, X_n). \tag{26}$$

Amplitudes involving several Casimir elements can be defined analogously. Let us study how such amplitudes depend on $x$. We first consider the simple example

$$\widehat{W}_2(C_2(x)) = \sum_a \Big( - \mathrm{Tr}\big(e_a \Psi^{-1} A \Psi e^a \Psi^{-1} A \Psi\big)(x) + \mathrm{Tr}\big(e_a \Psi^{-1} A \Psi\big)(x) \mathrm{Tr}\big(e^a \Psi^{-1} A \Psi\big)(x) \Big) \,. \tag{27}$$

The Casimir element, and the corresponding amplitudes, do not depend on the choice of the basis $\{e_a\}$ of $\mathfrak{g}$. Let us use the $x$-dependent basis $e_a = \Psi(x)^{-1} f_a \Psi(x)$, where $\{f_a\}$ is an arbitrary $x$-independent basis. This leads to

$$\widehat{W}_2(C_2(x)) = \sum_a \Big( - \mathrm{Tr}\,(f_a A(x) f^a A(x)) + \mathrm{Tr}\,(f_a A(x)) \mathrm{Tr}\,(f^a A(x)) \Big) \,. \tag{28}$$

This now depends on $x$ only through $A(x)$. Similarly, $\widehat{W}_{i+n}(C_i(x), X_1, \ldots, X_n)$ depends on $\Psi(x)$ and $\{e_a\}$ only through the combinations $M(x.e_a) = \Psi(x) e_a \Psi(x)^{-1}$, although these combinations may hide in expressions such as $K(x,x) e_a K(x, x_i)$. Therefore, using our $x$-dependent basis eliminates all dependence of $\widehat{W}_{i+n}(C_i(x), X_1, \ldots, X_n)$ on $\Psi(x)$. The only remaining dependence on $x$ is through the rational function $A(x)$.

Therefore, amplitudes that involve Casimir elements are rational functions of the corresponding variables. In particular they are functions on $\Sigma$ rather than on $\widetilde{\Sigma}$ as their definition would suggest, more specifically they are meromorphic functions with the same poles as $A(x)$. This is equivalent to the loop equations of matrix models [6], if we identify our amplitudes with the matrix models' correlation functions. For generalizations of this important result, see [7].

In order to ease the later comparison with conformal field theory, we will now tweak this result by changing the regularization of amplitudes at coinciding points. Instead of the regularization (12), let us use normal ordering, and define

$$\widehat{W}_{i+n}(C_i(x), X_1, \ldots, X_n) = \frac{1}{(2\pi i)^{n-1}} \sum_{a_1, \ldots, a_i} c_{a_1, \ldots, a_i}$$
$$\times \left( \prod_{i'=1}^{i-1} \oint_x \frac{dx_{i'}}{x_{i'} - x} \right) \widehat{W}_{i+n}(x_1.e^{a_1}, \ldots, x_{i-1}.e^{a_{i-1}}, x.e^{a_i}, X_1, \ldots, X_n) \,. \tag{29}$$

For example, let us compute $\widehat{W}_2(C_2(x))$. From the definition, we have

$$\widehat{W}_2(C_2(x)) = W_1(x.e_a) W_1(x.e^a) + \frac{1}{2\pi i} \oint_x \frac{dy}{(y-x)^3} \mathrm{Tr}\, M(y.e_a) M(x.e^a) \,. \tag{30}$$

We expand $M(y.e_a)$ near $y = x$ using $M' = [A, M]$ and $M'' = [A', M] + [A, [A, M]]$, and we find

$$\widehat{W}_2(C_2(x)) = \sum_a \Big( - \mathrm{Tr}\big(e_a \Psi^{-1} A \Psi e^a \Psi^{-1} A \Psi\big)(x) + \mathrm{Tr}\big(\Psi^{-1} A^2 \Psi e_a e^a\big)(x)$$
$$+ \mathrm{Tr}\big(e_a \Psi^{-1} A \Psi\big)(x) \mathrm{Tr}\big(e^a \Psi^{-1} A \Psi\big)(x) \Big) \,. \tag{31}$$

This differs from $\widehat{W}_2(C_2(x))$ (27) by one term. However, this difference does not affect the proof that the dependence on $x$ is rational, because the extra term still depends on $\Psi(x)$ and $\{e_a\}$ only through $M(x.e^a)$. More generally, amplitudes that involve the Casimir elements $C_i(x)$ are rational functions of their positions.

# 3 The Casimir W-algebra and its correlation functions

## 3.1 Casimir W-algebras and affine Lie algebras

For any Lie algebra $\mathfrak{g}$ and central charge $c \in \mathbb{C}$, there exists a W-algebra $\mathcal{W}_c(\mathfrak{g})$. If $\mathfrak{g}$ is simply laced and $c = \text{rank}\,\mathfrak{g}$, this algebra is a subalgebra of the universal enveloping algebra $U(\widehat{\mathfrak{g}}_1)$ of the level one affine Lie algebra $\widehat{\mathfrak{g}}_1$. The elements of $U(\widehat{\mathfrak{g}}_1)$ that corresponds to the generators of $\mathcal{W}_{\text{rank}\,\mathfrak{g}}(\mathfrak{g})$ can actually be written using the Casimir elements of $U(\mathfrak{g})$, and we call $\mathcal{W}_{\text{rank}\,\mathfrak{g}}(\mathfrak{g}) = \mathcal{W}(\mathfrak{g})$ the Casimir W-algebra associated to $\mathfrak{g}$. (See [3] for a review.)

The generators of the affine Lie algebra $\widehat{\mathfrak{g}}_1$ can be written as currents, i.e. as spin one chiral fields on $\Sigma$. Then the commutation relations of $\widehat{\mathfrak{g}}_1$ are equivalent to the operator product expansions (OPEs) of these currents. The currents are usually denoted as $J^a(x)$ where $x$ is a complex coordinate on the Riemann sphere and $a$ labels the elements of a basis $\{e^a\}$ of $\mathfrak{g}$, and their OPEs are

$$J^a(x_1)J^b(x_2) = \frac{\langle e^a, e^b \rangle}{x_{12}^2} + \frac{\sum_c \langle [e^a, e^b], e_c \rangle J^c(x_2)}{x_{12}} + O(1)\,. \tag{32}$$

Let us introduce the notation $J(x.e^a) = J^a(x)$. The OPEs then read

$$J(x_1.E_1)J(x_2.E_2) \underset{x_1 \to x_2}{=} \frac{1}{x_{12}^2} \langle E_1, E_2 \rangle + \frac{1}{x_{12}} J(x_2.[E_1, E_2]) + O(1)\,. \tag{33}$$

The Casimir subalgebra of $U(\widehat{\mathfrak{g}}_1)$ is generated by fields $\mathcal{W}^i$ that correspond to the Casimir elements $C_i$ (24) of $U(\mathfrak{g})$,

$$\mathcal{W}^i(x) = \sum_{a_1,\dots,a_i} c_{a_1,\dots,a_i}\Big(J(x.e^{a_1})\big(J(x.e^{a_2})\big(\cdots J(x.e^{a_i})\big)\big)\Big)\,, \tag{34}$$

where the large parentheses denote the normal ordering $(AB)(x) = \frac{1}{2\pi i} \oint_x \frac{dx'}{x'-x} A(x')B(x)$. We could define the fields $\mathcal{W}^i$ for arbitrary values of the central charge, but they form a subalgebra of $U(\widehat{\mathfrak{g}}_1)$ only if $c = \text{rank}\,\mathfrak{g}$ or $i = 2$. If $c \neq \text{rank}\,\mathfrak{g}$ and $i \neq 2$, their OPEs actually involve other fields (with derivatives of currents). There is some arbitrariness in the construction of our generators $\mathcal{W}^i$, starting with the choice of a basis of Casimir elements $C_i$. The quadratic Casimir $C_2$ is uniquely determined by the requirement that $T = \mathcal{W}^2$ generates a Virasoro algebra: then eq. (34) is called the Sugawara construction. Different choices of the Casimir elements $C_{i\geq 3}$ can lead to different properties of the fields $\mathcal{W}^i$, which in particular may or may not be primary with respect to $T$. This arbitrariness does not affect the property that $\mathcal{W}^i(x)$ depends meromorphically on $x$, which is all that we need.

We are interested in correlation functions $\left\langle \prod_{j=1}^N V_j(z_j) \right\rangle$ of $\mathcal{W}(\mathfrak{g})$-primary fields. Inserting $\mathcal{W}^i(x)$ in such a correlation function produces an $(N+1)$-point function which is meromorphic on $\Sigma$ as a function of $x$, with a pole of order $i$ at $x = z_j$. Equivalently, the OPE of $\mathcal{W}^i$ with the primary field $V_j$ is

$$\mathcal{W}^i(x)V_j(z_j) \underset{x \to z_j}{=} \frac{q_j^i}{(x-z_j)^i} V_j(z_j) + O\left(\frac{1}{(x-z_j)^{i-1}}\right)\,, \tag{35}$$

where the charge $q_j^i$ is assumed to be a known property of the field $V_j$. The OPE $JV_j$ is constrained, but not fully determined, by the OPEs $\mathcal{W}^i V_j$ and the definition of the fields $\mathcal{W}^i$.

One way to satisfy this constraint would to assume that the fields $V_j$ are affine primary fields, i.e. that there is a representation $\rho_j$ of $\mathfrak{g}$ on a space where $V_j$ lives, and that we have

the OPE i.e. $J(x.E)V_j(z_j) \underset{\pi(x)\to z_j}{=} \frac{\rho_j(E)}{\pi(x)-z_j}V_j(z_j) + O(1)$. In order to recover the OPE (35), the only constraints on the representation $\rho_j$ are then $\rho_j(C_i)V_j = q_j^i V_j$. However, in the presence of affine primary fields, the current $J$ would be meromorphic on $\Sigma$, and the symmetry algebra of our CFT would be the full affine Lie algebra $\widehat{\mathfrak{g}}_1$ rather than $\mathcal{W}(\mathfrak{g})$. We would therefore have too much symmetry, plus a large indeterminacy in the choice of the representations $\rho_j$.

We will now introduce a simpler assumption for the OPE $JV_j$, such that $J$ is not meromorphic on $\Sigma$, and lives on the bundle $\widehat{\Sigma}$ of Section 2.1. Then the symmetry algebra of our theory, which is generated by the meromorphic fields, will be only $\mathcal{W}(\mathfrak{g})$. This will in principle enable us to describe CFTs such as conformal Toda theories.

## 3.2 Free boson realization

Let $\mathfrak{g} = \mathfrak{h} \oplus \bigoplus_{r \in R} \mathfrak{g}_r$ be a root space decomposition of $\mathfrak{g}$, where $\mathfrak{h}$ is a Cartan subalgebra, $e^r \in \mathfrak{g}_r$, and $r^* \in \mathfrak{h}$. The W-algebra $\mathcal{W}_c(\mathfrak{g})$ has a free boson realization, i.e. a natural embedding into the universal enveloping algebra $U(\widehat{\mathfrak{h}})$. Now if $\mathfrak{g}$ is simply laced, then $U(\widehat{\mathfrak{h}})$ is not only a subalgebra, but also a coset of $U(\widehat{\mathfrak{g}}_1)$ by the nontrivial ideal $\mathcal{I}$ that is generated by relations of the type [8]

$$J(x.e^r) \propto \left( \exp \int J(x.r^*) \right),$$
(36)

where the large parentheses indicate that the exponential is normal-ordered. Modulo these relations, the Casimir W-algebra $\mathcal{W}(\mathfrak{g}) \subset U(\widehat{\mathfrak{g}}_1)$ coincides with $\mathcal{W}_{\text{rank}\,\mathfrak{g}}(\mathfrak{g}) \subset U(\widehat{\mathfrak{h}})$:

$$
\begin{array}{ccc}
\mathcal{W}(\mathfrak{g}) & \hookrightarrow & U(\widehat{\mathfrak{g}}_1) \\
\downarrow & & \Downarrow \\
U(\widehat{\mathfrak{h}}) & \simeq & U(\widehat{\mathfrak{g}}_1)/\mathcal{I}
\end{array}
$$
(37)

This diagram must commute modulo an automorphism of $\mathcal{W}(\mathfrak{g})$. In general, W-algebras have nontrivial automorphisms: for example, $\mathcal{W}(\mathfrak{sl}_n)$ has an automorphism that takes the simple form $\mathcal{W}^i \to (-1)^i \mathcal{W}^i$ for a particular definition of the generators $\mathcal{W}^i$. Correspondingly, specifying the OPEs between the generators $\mathcal{W}^i$ of $\mathcal{W}(\mathfrak{g})$ only determines the embedding $\mathcal{W}(\mathfrak{g}) \hookrightarrow U(\widehat{\mathfrak{g}}_1)$ (34) modulo an automorphism. We can therefore assume that the embedding is chosen so that the diagram commutes.

We will now use the natural embedding of $\mathcal{W}(\mathfrak{g})$ into $U(\widehat{\mathfrak{h}})$ for determining how $J(x.\mathfrak{h})$ behaves near a singularity, and then the relations (36) for determining how $J(x.\mathfrak{g})$ behaves. Given a singularity $z_j$, let us identify $U(\widehat{\mathfrak{h}})$ with the abelian affine Lie algebra that corresponds to the currents $(J(x.E))_{E \in \mathfrak{h}_j}$ for some Cartan subalgebra $\mathfrak{h}_j$. In order to reproduce the behaviour (35) of $\mathcal{W}^i(x)$, we assume that $V_j(z_j)$ is an affine primary field for that abelian affine Lie algebra,

$$J(x.E)V_j(z_j) \underset{x \to z_j}{=} \frac{\langle A_j, E \rangle}{x - z_j} V_j(z_j) + O(1) \quad, \quad (E \in \mathfrak{h}_j).$$
(38)

Here $A_j$ is an element of $\mathfrak{h}_j$ such that the leading term in the OPE (35) has the right coefficient $q_j^i$. The embedding of $\mathcal{W}(\mathfrak{g})$ into $U(\widehat{\mathfrak{h}})$ indeed induces a map from affine highest-weight representations of $U(\widehat{\mathfrak{h}})$ (with parameters $A \in \mathfrak{h}$) to highest-weight representations of $\mathcal{W}(\mathfrak{g})$ (with parameters $(q^i) \in \mathbb{C}^{\dim \mathfrak{h}}$); in terms of this map $A \to (q^i(A))$ we are requiring $q_j^i = q^i(A_j)$. Then let $\mathfrak{g} = \mathfrak{h}_j \oplus \bigoplus_{r \in R_j} \mathfrak{g}_r$ be the root space decomposition associated to the Cartan subalgebra $\mathfrak{h}_j$. The relations (36) imply

$$J(x.e^r)V_j(z_j) \underset{x \to z_j}{=} (x - z_j)^{r(A_j)} V_j(z_j) \times O(1).$$
(39)

Since the field $V_j(z_j)$ is an affine primary field for the abelian affine Lie algebra generated by the currents $(J(x.E))_{E\in\mathfrak{h}_j}$, it has a simple expression $V_j(z_j) = \exp\int^{z_j} J(x.A_j)$ in terms of the corresponding free bosons $(\int J(x.E))_{E\in\mathfrak{h}_j}$. The field $V_{j'}(z_{j'})$ is in general not an affine primary field for the same abelian affine Lie algebra, since in general $\mathfrak{h}_j \neq \mathfrak{h}_{j'}$. If we insisted on writing $V_{j'}(z_{j'})$ in terms of the free bosons $(\int J(x.E))_{E\in\mathfrak{h}_j}$, the resulting expression for $V_{j'}(z_{j'})$ would be complicated.

Let us interpret the behaviour of the currents $J(x.E)$ in terms of the representation of $\widehat{\mathfrak{g}}_1$ that corresponds to the field $V_j$. In the case $A_j = 0$, the OPE $J(x.E)V_j(z_j) = O(1)$ is trivial, and we simply have the identity representation. Nonzero values of $A_j$ can be reached from that case by the transformation

$$
\begin{cases}
J(x.E) & \to & J(x.E) + \frac{\langle A_j, E\rangle}{x-z_j} \,, & (E\in\mathfrak{h}_j)\,, \\
J(x.e^r) & \to & (x-z_j)^{r(A_j)} J(x.e^r)\,.
\end{cases}
\tag{40}
$$

This transformation can be interpreted as a spectral flow automorphism of the affine Lie algebra $\widehat{\mathfrak{g}}_1$, associated to the element $A_j \in \mathfrak{g}$. So $V_j$ belongs to the image of the identity representation under a spectral flow automorphism. This image is called a twisted module. For almost all values of $A_j$, the exponents $r(A_j)$ are not integer, and the corresponding twisted module is therefore not an affine highest-weight representation. For Lie algebras $\mathfrak{g}$ with Dynkin diagrams of the types $A_n$ and $D_n$, twisted modules have simple realizations in terms of free fermions [9].

## 3.3 Correlation functions as amplitudes

Let us show that the disconnected amplitudes of Section 2.2 are related to correlation functions of currents as

$$
\widehat{W}_n(X_1,\dots,X_n) = \frac{\langle J(X_1)\cdots J(X_n) V_1(z_1)\cdots V_N(z_N)\rangle}{\langle V_1(z_1)\cdots V_N(z_N)\rangle}\,,
\tag{41}
$$

provided the residues of the function $A$ in our Fuchsian system (1) coincide with the elements $A_j \in \mathfrak{g}$ that we introduced in eq. (38). By definition, correlation functions are determined by the OPEs and analytic properties of the involved fields. Therefore, to prove equation (41) it is enough to perform the following checks:

- Given the self-OPE of $J(X)$ (33), both sides of the equation have the same asymptotic behaviour (15) as $\pi(X_1) \to \pi(X_2)$.

- Given the OPE of $J(X)$ with $V_j(z_j)$ (38)-(39), both sides have the same asymptotic behaviour (23) as $\pi(X_1) \to z_j$.

- Since $J(X)$ is locally holomorphic (i.e. $\frac{\partial}{\partial \bar{x}} J(x.E) = 0$), both sides have the same analytic properties away from the singularities.

Let us be more specific on how the OPE (39) constrains $J(x.e^r)$. The coefficient of the leading term of this OPE is determined by eq. (36), and can be computed if needed. However, if $r(A_j) < -1$, this OPE also involves negative powers of $x-z_j$ whose coefficients are undetermined. These undetermined singular terms are not a problem, because the missing constraints are compensated by as many extra constraints on the regular terms of $J(x.e^{-r})$. Alternatively, we could prove eq. (41) in a region where $A_j$ is close enough to zero, and extend the result by analyticity in $A_j$.

As an additional check, notice that we have the following relation between correlation functions of a meromorphic field $\mathcal{W}^i(x)$ (35), and amplitudes that involve the corresponding

Casimir element (29),

$$\widehat{W_{i+n}}(\mathcal{C}_i(x), X_1, \ldots, X_n) = \frac{\langle \mathcal{W}^i(x)J(X_1)\cdots J(X_n)V_1(z_1)\cdots V_N(z_N)\rangle}{\langle V_1(z_1)\cdots V_N(z_N)\rangle} \ . \tag{42}$$

We have shown that the amplitude is a rational function of $x$, and this agrees with the properties of the field $\mathcal{W}^i(x)$.

By the correlation function $\langle V_1(z_1)\cdots V_N(z_N)\rangle$ we actually mean any linear combination of $N$-point conformal blocks, equivalently any solution of the corresponding $\mathcal{W}(\mathfrak{g})$-Ward identities. Let us review these solutions and their parametrization. Ward identities are linear equations that relate a correlation function of primary fields, and correlation functions of the corresponding descendent fields, which are obtained from primary fields by acting with the creation modes $\mathcal{W}^i_{-1}, \mathcal{W}^i_{-2}, \cdots$. Local Ward identities determine the action of the modes $\mathcal{W}^i_{n \leq -i}$, and $\mathcal{W}^i$ is left with $i-1$ undetermined creation modes, which appear as the residues of the poles of orders $1, \ldots, i-1$ in the OPEs $\mathcal{W}^i V_j$ (35). In total, each field $V_j$ is left with $\frac{1}{2}(\dim\mathfrak{g} - \mathrm{rank}\,\mathfrak{g})$ undetermined creation modes. (The identification of $L_{-1} = \mathcal{W}^2_{-1}$ with a $z$-derivative is irrelevant, as we do not know the $z$-dependence of our correlation function at this point.) The correlation functions are further constrained by the $\dim\mathfrak{g}$ global Ward identities, and the number of independent undetermined creation modes is

$$\mathcal{N}_{N,\mathfrak{g}} = \frac{1}{2}\Big((N-2)\dim\mathfrak{g} - N\,\mathrm{rank}\,\mathfrak{g}\Big) \ . \tag{43}$$

For example, in the case $\mathcal{N}_{4,\mathfrak{sl}_2} = 1$, we can choose the undetermined creation mode to be $L_{-1}$ acting on the last field, and Ward identities reduce arbitrary descendents to linear combinations of $\langle V_1(z_1)V_2(z_2)V_3(z_3)L^n_{-1}V_4(z_4)\rangle$ with $n \in \mathbb{N}$. Similarly, in the case $\mathcal{N}_{3,\mathfrak{sl}_3} = 1$, a basis of independent descendents is $\left\{\left\langle V_1(z_1)V_2(z_2)\big(\mathcal{W}^3_{-1}\big)^n V_3(z_3)\right\rangle\right\}_{n\in\mathbb{N}}$. A conformal block is an assignment of values for all elements of such a basis, equivalently a conformal block is an analytic function $f(\theta) = \left\langle V_1(z_1)V_2(z_2)e^{\theta\mathcal{W}^3_{-1}}V_3(z_3)\right\rangle$. Now a basis of a space of functions of $\mathcal{N}_{N,\mathfrak{g}}$ variables comes with $\mathcal{N}_{N,\mathfrak{g}}$ parameters. For example, in the case $\mathcal{N}_{4,\mathfrak{sl}_2} = 1$, there is a well-known basis called $s$-channel conformal blocks, whose elements are parametrized by the $s$-channel conformal dimension. For an example of a distinguished basis in the case $\mathcal{N}_{3,\mathfrak{sl}_3} = 1$ and in the limit $c \to \infty$, see [10].

Let us sketch how the conformal blocks' parameters are related to our function $A$. This function encodes the charges $q^i_j$ eq. (35), plus other parameters. Let us count how many. We have

$$A(x) = \sum_{j=1}^{N} \frac{A_j}{x - z_j} \quad \text{with} \quad \sum_{j=1}^{N} A_j = 0 \ , \tag{44}$$

as a consequence of $A(x) \underset{x\to\infty}{=} O(\frac{1}{x^2})$. Taking into account the invariance of amplitudes under conjugations of $A$ by elements of the Lie group $G$, we therefore have a total of $(N-2)\dim\mathfrak{g}$ parameters. Each field has $\mathrm{rank}\,\mathfrak{g}$ independent charges, for a total of $N\,\mathrm{rank}\,\mathfrak{g}$ charges. Therefore, the number of other parameters is $2\mathcal{N}_{N,\mathfrak{g}}$. From the case $\mathfrak{g} = \mathfrak{sl}_2$ [1], we expect that these parameters of $A$ are the $\mathcal{N}_{N,\mathfrak{g}}$ parameters of conformal blocks, plus $\mathcal{N}_{N,\mathfrak{g}}$ conjugate parameters. We will propose a geometrical interpretation of these parameters in Section 4.

# 4 Cycles of the bundle

## 4.1 Definition of cycles

Let an arc in $\widehat{\Sigma}$ be an equivalence class $\Gamma = [\gamma.E]$ under the action of $\pi_1(\Sigma)$, where $\gamma$ is an oriented arc in $\widetilde{\Sigma}$ and $E \in \mathfrak{g}$. To define the boundary of $\Gamma$, it would be natural to write

$\partial[(x,y).E] = [y.E] - [x.E]$. We will rather adopt the equivalent definition

$$\partial[(x,y).E] = \pi(y).M(y.E) - \pi(x).M(x.E) , \tag{45}$$

i.e. a formal linear combination of elements of $\Sigma \times \mathfrak{g}$, rather than $\widehat{\Sigma}$. In other words, this boundary is a $\mathfrak{g}$-valued divisor of $\Sigma$.

Let a cycle be a formal linear combination of homotopy classes of arcs, whose boundary is zero. For example, if $\gamma$ is an arc that starts at $x_0 \in \widetilde{\Sigma}$, such that $\pi(\gamma)$ is a closed loop, then using eq. (3) we find

$$\partial[\gamma.E] = x_0.M\big(x_0.(E - S_\gamma E S_\gamma^{-1})\big) . \tag{46}$$

Therefore, $[\gamma.E]$ is a cycle if and only if $E$ commutes with the monodromy $S_\gamma$ of $\Psi$ around $\gamma$. In particular, if $\gamma = \gamma_j$ is a small circle encircling $z_j$ and no other singularity, then using eq. (19) we see that $[\gamma_j.E]$ is a cycle if and only if $E \in \Psi_j^{-1}\mathfrak{h}_j\Psi_j$. But there exist many cycles that are not of that type.

We recall that the intersection number of two oriented arcs $\gamma, \gamma' \subset \Sigma$ is the integer

$$(\gamma, \gamma') = \sum_{x \in \gamma \cap \gamma'} (\gamma, \gamma')_x , \tag{47}$$

where $(\gamma, \gamma')_x$ is $+1$ (resp. $-1$) if the tangents of the two arcs at the intersection point $x$ form a basis with positive (resp. negative) orientation, so that $(\gamma, \gamma')_x = -(\gamma', \gamma)_x$. We define the intersection form of two arcs in $\widehat{\Sigma}$ as

$$(\Gamma, \Gamma') = \sum_{x = \pi(X) = \pi(X') \in \pi(\Gamma) \cap \pi(\Gamma')} (\pi(\Gamma), \pi(\Gamma'))_x \big\langle M(X), M(X') \big\rangle . \tag{48}$$

We have $(\Gamma, \Gamma') = -(\Gamma', \Gamma)$. The intersection is stable under homotopic deformations, and thus extends to linear combinations of homotopy classes. We consider two arcs as equivalent if they have the same intersection form with all cycles. Then the intersection form is non-degenerate, and is therefore a symplectic form on the space of cycles.

Let us consider an arc $[\delta_j.E] = [(x_0, z_j).E]$ that ends at a singularity $z_j$. Given the behaviour (18) of $M(X)$ near $z_j$, the boundary (45) of this arc makes sense only if $E$ commutes with the monodromy $S_j$ i.e. $E \in \Psi_j^{-1}\mathfrak{h}_j\Psi_j$. However, any element of $\mathfrak{g}$ has an orthogonal decomposition into an element that commutes with $S_j$, and an element of the type $E = F - S_j F S_j^{-1}$. With an element of this type, $[\delta_j.E]$ is equivalent to $[\gamma_j.F]$, where $\gamma_j$ is our arc around $z_j$. For example, $[\delta_j.E]$ and $[\gamma_j.F]$ have the same intersection with any arc $[\gamma'.E']$ such that $\gamma'$ passes between $x_0$ and $z_j$,

$$([\gamma_j.F], [\gamma'.E']) = \big\langle F, E' \big\rangle - \big\langle S_j F S_j^{-1}, E' \big\rangle = ([\delta_j.E], [\gamma'.E']) . \tag{49}$$

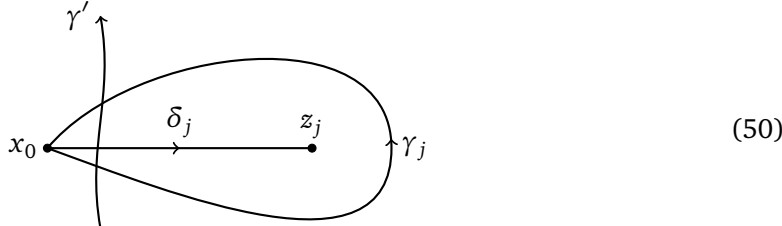

$$\tag{50}$$

Therefore, without loss of generality, we can assume that arcs that end at $z_j$ are of the type $[\delta_j.E]$ with $E \in \Psi_j^{-1}\mathfrak{h}_j\Psi_j$. We define generalized cycles to be combinations $\Gamma$ of arcs such that

$$\partial \Gamma \in \sum_{j=1}^{N} \Big[ z_j.\Psi_j^{-1}\mathfrak{h}_j\Psi_j \Big] . \tag{51}$$

## 4.2 Integrals of $W_1$ on cycles

Since the correlation functions $W_n$ live on $\widehat{\Sigma}$, they can be integrated on arcs and cycles of $\widehat{\Sigma}$. In particular, for any arc $\Gamma = [\gamma.E]$, we define the integral

$$\int_\Gamma W_1(X)dX = \int_\gamma W_1(x.E)dx . \tag{52}$$

A regularization is needed if the arc ends at a singularity $z_j$. If $E = \Psi_j^{-1}E_j\Psi_j$ with $E_j \in \mathfrak{h}_j$, we define

$$\int_{[(x_0,z_j).E]} W_1(X)dX = \int_{x_0}^{z_j} \left( W_1(x.E) - \frac{\langle A_j, E_j \rangle}{x - z_j} \right) dx - \langle A_j, E_j \rangle \log(x_0 - z_j) . \tag{53}$$

Since we have a symplectic intersection form, there exist symplectic bases of generalized cycles $\{A_\alpha, B_\alpha\}$, such that

$$(A_\alpha, A_\beta) = 0 \quad , \quad (B_\alpha, B_\beta) = 0 \quad , \quad (A_\alpha, B_\beta) = \delta_{\alpha\beta} . \tag{54}$$

We conjecture that, for any function $A$ and parameters $\{\theta_\alpha\}$ of conformal blocks, there exists a symplectic basis such that

$$\frac{1}{2\pi i} \int_{A_\alpha} W_1(X)dX = \theta_\alpha , \tag{55}$$

$$\frac{1}{2\pi i} \int_{B_\alpha} W_1(X)dX = \frac{\partial}{\partial \theta_\alpha} \log \langle V_1(z_1) \cdots V_N(z_N) \rangle . \tag{56}$$

That basis should respect the invariance of the correlation function $\langle V_1(z_1) \cdots V_N(z_N) \rangle$ under conjugations of $A$ with elements of $e^{\mathfrak{g}}$, and its independence from the choice of a solution $\Psi$ of eq. (2).

Let us evaluate the plausibility of our conjecture. To begin with, let us count cycles. For any singularity $z_j$, the cycles $[\gamma_j.E]$ with $E = \Psi_j^{-1}E_j\Psi_j \in \Psi_j^{-1}\mathfrak{h}_j\Psi_j$ are the $A$-cycles that correspond to the components of $A_j$ along the Cartan subalgebra $\mathfrak{h}_j$,

$$\frac{1}{2\pi i} \int_{[\gamma_j.\Psi_j^{-1}E_j\Psi_j]} W_1(X)dX = \langle E_j, A_j \rangle , \qquad (E_j \in \mathfrak{h}_j) . \tag{57}$$

Then the corresponding $B$-cycles must include terms of the type $[\delta_j.E]$. Such cycles account for the $N$ rank $\mathfrak{g}$ parameters that are equivalent to the charges $q_j^i$ of the fields $V_j(z_j)$. Let us determine the dimension of the space of the rest of the cycles. Let us build these cycles from $N$ loops with the same origin $x_0$, with each loop going around one singularity:

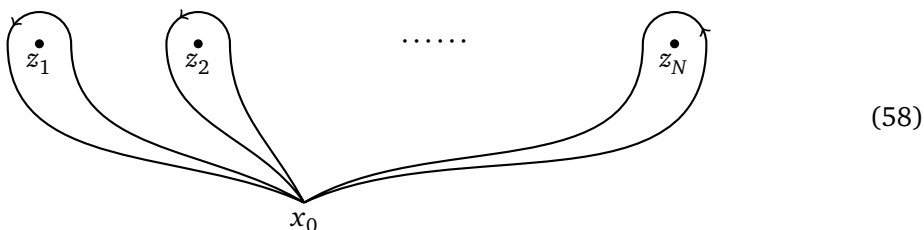

$$\tag{58}$$

The resulting combination of arcs belongs to $\sum_{j=1}^N [\gamma_j.\mathfrak{g}]$, with a boundary in $[x_0.\mathfrak{g}]$. Requiring that the boundary vanishes, and imposing the topological relation $\sum_{j=1}^N \gamma_j = 0$, we have

$(N-2)\dim\mathfrak{g}$ independent cycles. After subtracting our $N$ rank $\mathfrak{g}$ A-cycles around the singularities, the number of independent cycles is twice the number $\mathcal{N}_{N,\mathfrak{g}}$ (43) of parameters of conformal blocks, in agreement with our conjecture.

Next, let us test our conjecture in the case $\mathfrak{g} = \mathfrak{sl}_2$ and $N = 3$. In this case, the correlation function is the explicitly known Liouville three-point function at $c = 1$, and we have found a suitable basis [2]. However, this basis is ad hoc and cannot easily be generalized.

Let us compare our conjecture with the more general results of Bertola [11] on isomonodromic tau functions. To a derivation $\delta$, for instance $\delta = \frac{\partial}{\partial\theta_\alpha}$, we will now associate a cycle $B_\delta$. We choose an oriented graph $\Gamma \subset \Sigma$ such that $\Sigma - \Gamma$ is simply connected:

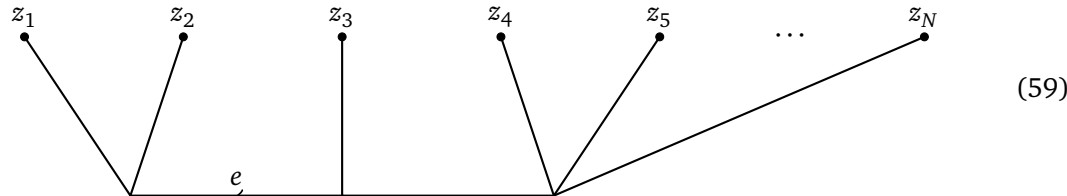

$$(59)$$

Given a representative $\Sigma_0 \subset \widetilde{\Sigma}$ of $\Sigma - \Gamma$, we define

$$B_\delta = \sum_{e \in \Gamma} \left[ e.\delta S_e S_e^{-1} \right] ,\qquad (60)$$

where $e$ is an edge of $\Gamma$ when accessed from the right in $\Sigma_0$, and $S_e$ is the monodromy from the right to the left of $e$ in $\Sigma_0$. Then $B_\delta$ is a generalized cycle in the sense of eq. (51). Moreover, in our notations, Malgrange's form and its exterior derivative can be written as

$$\omega(\delta) = \frac{1}{2\pi i} \int_{B_\delta} W_1(X) dX \qquad , \qquad d\omega(\delta_1, \delta_2) = (B_{\delta_1}, B_{\delta_2}) . \qquad (61)$$

This suggests that our formalism is well-suited for dealing with Malgrange's form. However, this also shows that we cannot have $B_\alpha = B_{\frac{\partial}{\partial\theta_\alpha}}$ as one might naively have expected, because Malgrange's form is not closed, and differs from the logarithmic differential of the tau function by a term that does not depend on the positions $z_j$ of the poles [11]. Therefore, the cycle $B_{\frac{\partial}{\partial\theta_\alpha}}$ might be an important term of the eventual cycle $B_\alpha$.

# 5 Conclusion

Our main technical result is the construction of the bundle $\widehat{\Sigma}$, which we believe describes the intrinsic geometry of a Fuchsian system. This in principle allows us to avoid splitting the surface $\Sigma$ in patches, with jump matrices at boundaries, as is otherwise done for describing solutions of the Fuchsian system [11]. Our construction also works for more general differential systems, in particular if $A(x)$ has poles of arbitrary order rather than the first-order poles that we need in conformal field theory. The important assumption is that $A(x)$ is meromorphic.

In our application to conformal field theory with a W-algebra symmetry, our construction is particularly useful for computing correlation functions that involve currents. The application to correlation functions of primary fields, i.e. to the tau functions of the corresponding integrable systems, is still conjectural.

Let us consider the classical limit of the Fuchsian system (2), $\frac{\partial}{\partial x} \to \epsilon \frac{\partial}{\partial x}$ and then $\epsilon \to 0$. This limit can be defined so that our bundle $\widehat{\Sigma}$ tends to a Riemann surface, namely a $|\text{Weyl}(\mathfrak{g})|$-fold cover of $\Sigma$ called the cameral cover associated to $A$ [12]. The cameral cover can be understood as containing as much information as all the spectral curves $\Sigma_\rho = \{(x, y) \in T^*\Sigma | \det_\rho(y - A(x)) = 0\}$, for any representation $\rho$ of $\mathfrak{g}$.

## Acknowledgments

We are grateful to Taro Kimura for collaboration at early stages of this work. We thank Leonid Chekhov, Jorgen Andersen, Gaëtan Borot, Oleg Lisovyy, Pasha Gavrylenko, and Ivan Kostov for discussions. We are grateful to Pasha Gavrylenko and Oleg Lisovyy for helpful comments on the draft of this text. We also thank the anonymous SciPost reviewers (and especially the second reviewer) for their valuable suggestions. BE was supported by the ERC Starting Grant no. 335739 "Quantum fields and knot homologies" funded by the European Research Council under the European Union's Seventh Framework Programme. BE is also partly supported by the ANR grant Quantact : ANR-16-CE40-0017.

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
