# Peer review of "The geometry of Casimir W-algebras"

_SciPost Physics, doi:SciPost Phys. 5, 051 (2018)_

## Round 2 · Referee Report · Anonymous (Referee 1) · 2018-2-15

Strengths

  1. The authors succeed in their goal to develop a generalisation to an arbitrary (simply-laced) Lie algebra of their earlier work (in the context of $gl_2$) on the interesting relationship between Fuchsian systems and conformal field theory, to explain the mathematics behind it, then to use it to calculate correlation functions that involve currents within certain $W(g)$ symmetric conformal field theories and discuss their properties. It seems to me this is a positive mathematical development that is likely to be elaborated further, possibly to evaluate correlation functions of primary fields, or to extend its scope.

Weaknesses

  1. The paper fails to cite properly the work of others in the sense that the data provided with the references is incomplete compared to standard conventions. For example, Ref 8 is referred to as '1997 book', while most other papers are referred to only by their arxiv numbers (the exception being a PhD thesis that does not have an arxiv number). According to the instructions for authors, the DOI numbers should also be included for published papers yet none are provided in this article.

Report

While the paper is inspired by ideas from physics (since conformal field theories have their origins there), and is a continuation of earlier work by the authors and others, the focus is on the mathematics, developing ideas and techniques without any clear indication (it seems to me) concerning where they might be useful in a physical context. For this reason it is likely to be of interest to a relatively small group of mathematical physicists rather than more broadly to the readership of a journal devoted to physics. Nevertheless, as this is quite common, the ideas, techniques and results reported should be published as interesting steps within a larger program of work.

Requested changes

Improve the data provided alongside the references (ie add the publication data, journal etc, and also DOIs for the purposes of creating external links to journals).

Consider adding some explanation as to why the investigation ought to be of interest to physicists.

---

## Round 2 · Referee Report · Anonymous (Referee 2) · 2018-3-7

Strengths

  1. The paper is an interesting attempt to provide a geometric framework for new ways to consider W-algebra conformal blocks and correlation functions

  2. It is likely to generate further work to understand this area

Weaknesses

  1. There are several places where the arguments are not clear or explained well.

Report

This looks an interesting paper, I think it is likely to stimulate further work in this active area, but there are several places where I could not follow the arguments or where they were unexpected and not explained. I think that if these were addressed it would be considerably improved, from my point of view. I'll make a list of places where I thought it was not clear in the detailed changes, but two points, one possibly important, one just historical, I would like to mention here.

It is from one point of view very surprising that, as in reference [3], the conformal blocks depend on so few parameters. From simple W-algebra identities one would expect a conformal block to have an infinite number of parameters which could be reduced if the fields involved were suitably degenerate, but this does not appear to happen here. I would have expected a comment on this fact - if it is well understood by now, then a reference would have been good.

I was a little surprised that the authors decided to choose the name "Casimir W-algebra" for the specialisation of what is already known as a Casimir W-algebra to the value of $c$=rank$\mathfrak g$ - perhaps they were unaware that this term had already been used, e.g. in arXiv:hep-th/9404113

Requested changes

These are in order of occurrence in the paper. Some requests are just that the authors consider changing their presentation, others are really requests for more clarity that I think are needed (from my point of view)

  1. Section 2.3, there is no discussion of the choice of $C_i$, whether a basis is chosen such that the fields $W_i$ are Virasoro primary or not. I could not easily see if this was needed or not in what follows - a comment on this would be helpful.

  2. Section 3.1 - the name "Casimir W-algebra" is already used in the literature to describe the algebra ${\cal W}_c(\mathfrak g)$, I can see that the authors would like to give their particular restriction to $c=rank(\mathfrak g)$ a name, but perhaps they could choose one that is not already used, or at least refer to this so that readers who are familiar with the older literature do not get confused?

  3. On page 7, it is not clear which representations of $\mathfrak g$ are allowed if $V_j$ are to be primary fields for $\mathfrak g$. If they are the unitary or integrable representations of $\mathfrak g$, then clearly this is too few; if they are more general, then they will not be representations of the vertex algebra corresponding to $\mathfrak g$ and so possibly some assumptions from CFT would fail. I know that the authors say that this is not what they plan to do, but it is still not clear what they are saying they are not going to do.

  4. On page 7, I am not sure what the authors mean by "This diagram commutes only up to multiplication with scalar factors". If they mean that restricting the fields from $\mathfrak g$ to $\mathfrak h$, it is not the case that for higher $W_i$ fields the relation is just a scalar factor - the restriction of the tensor $C_i$ to the fields in $\mathfrak h$ which typically not be equal to the corresponding field when summed over all the fields in $\mathfrak g$, up to a factor. This can be easily seen since the requirement of primality is that the tensor is traceless, and the requirement that an invariant tensor of rank 4 for $sl(n)$ is traceless when summed over all indices does not lead to it being traceless when summed over only the Cartan indices. If they actually mean using the maps then the relations in $\cal I$ should mean that (since the the energy-momentum tensor is unique) the two expressions given are in fact the same (modulo $\cal I$), for example for $sl(2)$, with the relations in $\cal I$ being $J^\pm = \exp(\pm i\sqrt 2 \varphi)$, we get $ (J^+ J^- + J^- J^+) = 2 (i \partial\varphi)^2$, and so we end up with equality of the two expressions (modulo $\cal I$).

  5. On page 8, equation (3.8) only fixes the OPEs of the fields in $\mathfrak h_j$, yet the sum in (3.7) appears to be over all $\mathfrak g$. Is the sum in (3.7) over only the CSA? If so, which CSA? This is rather confusing - since the tensor $C_i$ is group invariant, it should not matter to which CSA one restricts, but the restriction should be done after the choice of CSA. It appears in the paper that once can use the full expression, summing over $\mathfrak g$ in (3.7) and then restrict to any particular $\mathfrak h_j$ and still use the same expression. One has $W_i = \sum_{a \in\mathfrak g} C_{a_1,\ldots a_i} (J^{a_1} ( \ldots ( J^{a_m})..) = \sum_{\alpha \in \mathfrak h} C'_{\alpha_1\ldots \alpha_i} (J^{\alpha_1} ( \ldots ( J^{\alpha_m})..)$ but $C_{\alpha_1\ldots \alpha_i} \neq A C'_{a_1,\ldots a_i} $ for some number $A$, in general, and so the relation between (3.7), (3.8) and (3.9) needs to be clarified.

  6. On page 8, I did not easily see why $V_{j'}(z')$ could not be expressed using the same free boson as $V_j(z_j)$ if the intention is only to replicate the W-algebra primary fields. They may not be expressed as simple exponential, but if $A_j$ and $A_{j'}$ define different Cartan subalgebra, then could one not find the conjugation required to take $A_{j'}$ to be in the $\mathfrak h_j$ using the generators $\mathfrak g_0$ expressed in terms of the bosons in $\mathfrak h_j$ in and similarly conjugate the primary field? The resulting expression will be in terms of the free bosons of $\mathfrak h_j$ - the expressions may be ugly, but wouldn't $V_{j'}$ will still be expressed in terms of those free bosons?

  7. On page 8, two lines above section 3.3, it says "These representations" - I was not sure which representations were being refered to .

  8. This is surely my ignorance, but in section 3.3, why is only the leading behaviour as $\pi(X_1)\to \pi(X_2)$ needed? If $r(A_j)$ in (3.9) is large and negative, are not all the singular terms important?

  9. On page 9, why does each field $V_j$ come with only $1/2 (dim(\mathfrak g) - rank(\mathfrak g))$ undetermined descendents? I would have said that for a W-algebra, all the descendents in the OPE were undetermined from the W-algebra commutation relations, it is only for the Virasoro algebra that they are determined by the descent relations (from BPZ). If instead the authors mean there are $1/2 (dim(\mathfrak g) - rank(\mathfrak g))$ undetermined parameters, which appears to be what they mean given the parameter counting in (3.12), why is this? I thought this was one of the surprising results of [3] which I took to be an assumption in this paper. If it is explained elsewhere why the number of parameters is fixed, a reference would be most helpful.

  10. On page 11, it says "We conjecture is that".

---

## Round 3 · Referee Report · Anonymous (Referee 1) · 2018-6-28

Report

The authors have made a change I suggested (additional motivation for their work) but have not adjusted the referencing. Instead, they have provided a sentence of justification for their policy of referring only to arxiv preprints. If their justification is acceptable to SciPost (despite the fact it appears to be at variance with the SciPost instructions with regard to references), then I will not insist on any changes. Though I have to say I agree with the SciPost policy in this regard - I would prefer to see the published journal reference together with the arxiv post and then be able to decide for myself, if necessary, whether changes to a published version are significant - moreover, in their parenthetic remark, the authors do not object to it either provided SciPost does the job of supplying the missing data!

I notice that they have also made some corrections in response to the other referee.

---

## Round 3 · Referee Report · Anonymous (Referee 2) · 2018-7-13

Strengths

  1. The paper is an interesting attempt to provide a geometric framework for new ways to consider W-algebra conformal blocks and correlation functions

  2. It is likely to generate further work to understand this area

Weaknesses

  1. There are several places where the arguments are not clear or explained well. See the report for details

Report

1. I find it hard to agree with the counting of parameters in W-algebra conformal blocks. The authors give two examples where the counting ${\cal N}_{N,g}=1$, which are for $N=4$, $g=sl(2)$ and for $N=3$, $g=sl(3)$. For the first case, I agree that this counts the number of parameters needed to fix the conformal block. The four point block on primary fields is given by

$$ \langle h | \phi_{h'}(1) \phi_{h''}(z) | h'''\rangle = z^{H - h''-h'''}(1 + z \frac{(H +h-h')(H + h''-h''')}{2H} + \ldots) $$
which is fixed, for given $z$, by determining H, and the four-point block of any Virasoro descendants is also fixed uniquely once $H$ is fixed.

This is very different to the case of the W3 algebra 3-point block where the three point functions of any W3 descendants are fixed up to the infinite set of parameters

$$\langle h,w| \phi_{h',w'}(z) (W_{-1})^n |h'',w''\rangle = c_n z^{h-h'-h''-n}$$
It is true that this is equivalent to a single undetermined function defined e.g. as $F(u) = \sum c_n u^n$, but I really think that this is not the same as the block being determined up to one parameter. One parameter means one real number, not an infinite set of real numbers.

It is also a little odd to say that there are $1/2(dim(g) - rank(g))$ ''undetermined descendants'' in the OPEs ${\cal W}^i V_j$ : for $sl(2)$ this is just 1 and the ''undetermined descendant'' which is the pole of order 1 in the OPE

$$ T(z) V_j(w) = \frac{ h_j V_j(w)}{(z-w)^2} + \frac{ \partial V_j(w)}{(z-w)} + O(1) $$
is exactly the derivative field. Saying that this is undetermined is the same as saying that one does not understand the complex structure of the conformal block. Perhaps this is related to what ${\cal N}_{N,g}$ counts - but it is not explained.

The example of the 4 point block of the Virasoro algebra being determined up to one parameter makes it clear that the block (as a function of $z$) is fully determined once this one single real number is known. This is not the same for the 3-pt block of the W3 algebra.

$$] I wonder if the authors really mean that the Virasoro four point function is undetermined up to a FUNCTION of $H$, e.g. $c(H)$ as might appear below [ \langle h | \phi_{h'}(1) \phi_{h''}(z) | h'''\rangle = \int {\mathrm d} H \; c(H)\;F(h,h',h'',h''';H;z) $$
where $F(h,h',h'',h''';H;z)$ is the conformal block depending on the one parameter $H$.[]

The counting ${\cal N}_{N,g}$ clearly counts something, but it is not the number of parameters appearing in a block if one only uses W-algebra Ward-identities.

If one can also use identities resulting from the construction in terms of the affine algebra at level 1, then this may be the case, I do not know, but that is not what is stated here.

$$] I think this point is sufficiently important that the paper should not be published as it is without a clearly elucidation. If one real number (the weight of the intermediate channel in a Virasoro 4-pt block) is "one parameter", then an arbitrary function of one variable is not "one parameter". It could be that ${\cal N}_{N,g}$ counts the number of undetermined functions appearing in the correlator [as opposed to the block]. It could just be that the "explanation" added after (3.11) is wrong. I really would like to see a clear explanation. [$$
2. The authors seem to have misunderstood my previous comment on the diagram (3.6). What I was saying was that this diagram should actually commute, for a suitable definition of the maps; it was the previous statement that it only commuted up to a scalar factor that was wrong. To include this diagram and then to say that This diagram is in general not commutative'' is not a helpful statement to make. \[\] 3. The normal ordering implicitly defined through the regularised kernels (2.11) does not seem to be the same as that defined in (3.3). I am not sure that the Casimir algebras as described in this paper can be constructed through the correlation functions defined by (2.12), (2.13). To reproduce (3.3), one would need to add derivative fields at coincident points which is not discussed here. \[\] 4. Having re-read section 3.2, I have started to worry whether the requirement of cocycles for the free-field construction of $g$ at level 1 will play a role. They are not mentioned explicitly here, being hidden in the $\sim$ in equation (3.5), but they do complicate the construction of the affine fields and I do not see why they are also not needed in the construction of the fields $V_j(z_j)$ just after (3.8). I also think the statement thattwisted modules differ ... except in the case $A_j=0$'' is wrong. The twisted modules'' will be the same as standard highest weight modules whenever $A_j$ is in the weight lattice of $g$. For example, for $g=sl(2)$, with the simple root $\alpha=(\sqrt 2)$, then the field $exp( \int J(z.(1/\sqrt 2)))$ is just the field correpsonding to the highest weight of the non-trivial representations of spin $s=1/2$, and the field $exp( \int J(z.(\sqrt 2)))$ is just the field $J^+(z)$, which is in the vacuum representation. I would also point out that reference [9] only deals with $A_n$ and $D_n$ (of the simply-laced algebras), not $E_n$, and so it is possibly an open question whether one can construct a twisted module for all simply-laced affine algebras using free fermions. There is atranscendental'' free fermion construction for $E_8$ but the methods of [9] would not appear to be easy to apply.

Requested changes

1- to explain clearly how ${\cal N}_{N,g}$ does, or does not, count parameters or unknown functions or whatever.

2- to clear up when the diagram (3.6) is commutative

3- to comment on the normal ordering implied by (2.12), (2.13)

4- to clear up when the "twisted" fields they define are actually twisted and when not.

---

## Round 4 · Referee Report · Anonymous · 2018-10-12

Report

As this is my fourth review, I will only say that I think the authors are not likely to agree on precisely the meaning of the number of parameters parametrising a space of functions. Given the fact that there is a continuous map from $\mathbb{R}^2$ to $\mathbb{R}$, I think this is a rather unclear concept without appropriate extra conditions of smoothness, but I think this is not sufficient to stop me recommending the paper. the authors have a clear point of view which I think should be heard. I personally also question the statement, or requirement, that the function f($\theta$) at the bottom of page 10 is analytic, since the coefficients can be freely chosen. When one of the representations has a null vector this seems likely to be the case, but in general I think it would have to be an extra assumption. Still, this is a very minor point which I hope will not confuse the reader. I am happy to recommend publication of the article as it stands. I am very grateful to the authors for all the changes they have made in response to my comments which has helped me understand the paper.

---

## Round 4 · Author Response

We are grateful to the reviewer for this report, which allowed us to correct several technical mistakes and imprecisions. (This does not affect our conclusions.)

---

## Round 4 · List of Changes

Reply to Report 2 on revised version:

1. We have tried to improve the explanations, with a more detailed and explicit elaboration of our two examples. Basically, one should not confuse coefficients, variables and parameters: a basis of a space of functions of one variable is a one-parameter family of functions; a function in that space is a linear combination of basis functions, with infinitely many coefficients.

2. We now state that the diagram is indeed commutative, provided an embedding is appropriately chosen.

3. The reviewer rightly points out that the regularization in (3.3) differs from the regularization of amplitudes at coinciding points. Strictly speaking, this does not affect our proof of the relation between correlation functions and amplitudes in Section 3.3. Nevertheless, the third bullet item in that proof implicitly suggested a relation between amplitudes of Casimir elements, and correlation functions of generators of the Casimir algebra: this was wrong, and unnecessary for the proof. In order to clarify this point, we have added a paragraph at the end of Section 2.3, where we give an alternative definition of amplitudes of Casimir elements, using normal ordering rather than our earlier regularization. This leads to the new eq. (3.11), which provides an extra check of the relation between correlation functions and amplitudes.

4a. We agree that cocyles are hidden in the $\propto$ sign in (3.5) and in the $O(1)$ factor in (3.8). We prefer leaving the cocycles hidden as they are not essential to our argument. The reference [8] that we cite gives the correct cocycles.

4b. Our statement that twisted modules are affine highest-weight representations only if $A_j=0$ was indeed too strong. We have corrected it. We have however refrained from discussing the precise conditions for twisted modules to be affine highest-weight representations, as we do not need it.

4c. We have added the restriction of [9]'s free fermion construction to the A- and D-series.

---

## Editorial Decision

published